# Addressing Inflammaging and Disease-Related Malnutrition: Adequacy of Oral Nutritional Supplements in Clinical Care

**DOI:** 10.3390/nu16234141

**Published:** 2024-11-29

**Authors:** Nagaia Madini, Alessandra Vincenti, Alice Beretta, Sara Santero, Giulia Viroli, Hellas Cena

**Affiliations:** 1Laboratory of Dietetics and Clinical Nutrition, Department of Public Health, Experimental and Forensic Medicine, University of Pavia, 27100 Pavia, Italy; alessandra.vincenti@unipv.it (A.V.); alice.beretta@unipv.it (A.B.); sara.santero@unipv.it (S.S.); giulia.viroli@unipv.it (G.V.); hellas.cena@unipv.it (H.C.); 2Clinical Nutrition Unit, ICS Maugeri IRCCS, 27100 Pavia, Italy

**Keywords:** malnutrition, inflammation, supplements, ONS, FSMPs

## Abstract

Background: Disease-related malnutrition, with or without inflammation, in older adults is currently emerging as a public health priority. The use of Foods for Special Medical Purposes, including Oral Nutritional Supplements, and supplements is crucial to support patients in achieving their nutritional needs. Therefore, this article aims to comprehensively provide an analysis of the adequacy of FSMPs in meeting the nutritional requirements of different age-related diseases and takes into account the emerging role of inflammation. Moreover, it provides an identikit of the ideal products, following the pathology-specific guidelines. Methods: Data on 132 products were gathered through face-to-face meetings with companies’ consultants. Specifically, information on energy, macronutrient, and micronutrient contents were collected, as well as on texture and flavors, osmolarity, cost, and packaging. Results: Most FSMPs met the daily recommendations for energy and protein intake. Nonetheless, few products contained β-hydroxy-β-methylbutyrate, optimal Branched-Chain Amino Acids ratios, arginine, glutamine, and omega-3 fatty acids. Furthermore, a marked predominance of FSMPs with a high osmolarity (85.7%), sweet taste (72%), and only animal protein content (79.5%) was observed. Cost analysis of FSMPs revealed a mean cost of EUR 5.35/portion. Products were mostly adequate for cancer, neurodegenerative diseases, diabetes, inflammatory bowel disease, end-stage kidney disease, dysphagiam and chronic obstructive pulmonary disease. However, gaps have been found for sarcopenia and abdominal surgery. Conclusion: In light of the current market landscape, there is a need for a comprehensive regulation that indicates the optimal composition of FSMPs and the production of such products to tackle disease-related malnutrition.

## 1. Introduction

Aging is a complex physiological process influenced by genetic, lifestyle, social, and environmental factors [1]. Over recent decades, research in geroscience has identified several interconnected hallmarks for aging including inflammation, cellular senescence, stem cell exhaustion, and mitochondrial dysfunctions [2].

While inflammation is a crucial component of the immune response, its chronic presence, known as “inflammaging” [3], poses significant risks for morbidity and mortality in the elderly [4,5].

Evidence suggests that systemic inflammation increases with age, marked by elevated levels of proinflammatory cytokines such as tumor necrosis factor-alpha (TNF-α), interleukin (IL)-6, IL-1, and C-reactive protein (CRP) has been established [5,6]. These biomarkers are linked to decline in muscle strength, compromised bone metabolism, and impairment of nutritional status, contributing to age-related diseases such as sarcopenia, cardiovascular diseases, neurodegenerative disorders, type 2 diabetes, arthritis, cancer, and osteoporosis [7,8,9,10,11]. Moreover, inflammation is a recognized etiologic criterion for the diagnosis of malnutrition, as highlighted by the European Society of Clinical Nutrition and Metabolism (ESPEN) and the Global Leadership Initiative on Malnutrition (GLIM) [12].

Malnutrition, particularly protein-energy malnutrition (PEM), is highly prevalent among older adults, affecting 0.8% to 11% in European countries, with Italy at the higher end of this rate (11%) [13]. The risk of developing malnutrition in the elderly ranges from 23% to 53% [14,15], and these figures are expected to rise as the global population ages [16,17].

Growing concerns about the adverse health impacts and high prevalence of both malnutrition and inflammation have led to calls for early-stage interventions [18,19,20].

One promising treatment approach involves the use of Foods for Special Medical Purposes (FSMPs) [21], including Oral Nutritional Supplements (ONSs). FSMPs are specially formulated to meet the nutritional needs of patients with various medical conditions [20]. The European Society for Clinical Nutrition and Metabolism (ESPEN) endorses the use of oral nutritional supplements (ONSs) in older adults with malnutrition or at risk of malnutrition when dietary counseling and food fortification alone are not sufficient to increase dietary intake and reach nutritional goals [20,22]. Offering ONSs to these patients has been shown to improve dietary intake and body weight, reduce the risk of complications and hospital readmissions, and mitigate functional decline post-discharge [20].

Additionally, in polymorbid medical patients who are malnourished or at risk, ONS prescriptions are recommended to improve nutritional status, specifically maintaining muscle mass, while also improving quality of life and overall survival [20,23]. Despite the extensive evidence supporting their benefits for malnourished patients [24,25,26,27,28], research on their role in treating inflammaging remains limited [29]. Furthermore, there is a lack of well-defined legislative definitions and regulations governing their use for different diseases, particularly in terms of compositions tailored to specific disease-related needs. Given this gap, the purpose of this narrative review is to (i) provide a comprehensive overview of the current Italian market landscape of FSMPs (including ONSs) and supplements (including fortified foods), (ii) assess their adequacy in meeting nutritional requirements of different age-related diseases, (iii) evaluate new perspectives on the role of FSMPs in treating inflammaging, and (iv) provide a preliminary identikit of ideal products for malnourished patients with different diseases.

### 1.1. Malnutrition Classification According to ESPEN

Malnutrition is a broad term encompassing all deviant nutritional states. According to the ESPEN, malnutrition can be divided into three primary categories: malnutrition related to inflammatory disease, malnutrition related to non-inflammatory diseases, and malnutrition related to starvation [20].

#### 1.1.1. Disease-Related Malnutrition with Inflammation

This category includes chronic diseases with mild to moderate inflammation. It is essential to recognize that inflammation in these conditions may regress, recur, or be exacerbated, depending on the disease course, treatment modalities, and overlapping events or complications. Acute conditions or chronic diseases complicated by moderate acute exacerbations, like the active phase in Crohn’s disease, also fall under this category. Therefore, some diseases (e.g., Chronic Obstructive Pulmonary Disease (COPD)) can be present in both severe-moderate and mild-moderate inflammatory conditions.

#### 1.1.2. Disease-Related Malnutrition Without Inflammation

Cachexia, a multifactorial syndrome characterized by a continuous loss of skeletal muscle mass (with or without loss of fat mass) falls under this category. Cachexia cannot be completely reversed by conventional nutritional support and leads to progressive functional impairment. While hypercatabolism is a key factor in cachexia, it can exist without overt systemic inflammation [30,31].

#### 1.1.3. Starvation-Related Malnutrition

PEM is the phenotypic expression of starvation-related malnutrition. This condition is dictated by food insecurity, including poverty, famine, and war, and lacks clear or perceptible inflammatory components [12].

#### 1.1.4. Other Specific Conditions

For age-related body composition changes, malnutrition has also been defined in terms of sarcopenia and frailty. Sarcopenia is a progressive and generalized musculoskeletal disorder associated with an increased likelihood of adverse outcomes, including falls, fractures, physical disability, and mortality.

Frailty is a clinical state characterized by increased vulnerability due to a decline in physiological reserve and function across multiple organ systems associated with aging.

Moreover, various conditions, such as psychiatric diagnoses (e.g., eating disorders and depression), malabsorption, obstruction or dysmotility conditions, dysphagia, and neurodegenerative diseases (e.g., Alzheimer’s, Parkinson’s, and Amyotrophic Lateral Sclerosis), can lead to malnutrition. These conditions are not clearly associated with inflammation, but still contribute significantly to nutritional decline.

This overview highlights the multifaceted nature of malnutrition and the importance of recognizing the different etiologies to provide appropriate treatment. The classification system proposed by ESPEN serves as a framework for understanding the diverse causes of malnutrition and underscores the need for tailored interventions, including the use of FSMPs, ONSs, and supplements, to address specific nutritional needs.

### 1.2. FSMPs, ONSs, and Supplements

Given the health-related consequences of malnutrition, ONSs were developed as nutrient-dense products designed to meet the total or partial nutrient requirements of patients with a limited, impaired, or disturbed capacity to ingest, digest, absorb, metabolize, or excrete ordinary food or certain nutrients. These supplements can also cater to patients with specific medically determined nutrient requirements [32].

Country-specific guidelines regulate the use of ONSs. In Europe, ONSs are considered FSMPs, meaning these foods are specially formulated and designed for the dietary management of patients with malnutrition and should be used only under medical supervision [32]. Although all ONSs are FSMPs, the latter are a broader category that includes different medical foods formulated for the dietary management of patients suffering from disorders, diseases, or medical conditions that result in nutritional vulnerability, that is, the inability or severe difficulty in feeding themselves using common foods, including food supplements, to meet their nutritional needs [32].

To accommodate personalized nutritional needs, ONSs come in a variety of formats (e.g., liquid, powder, pudding, pre-thickened), volumes, flavors [20,33], and compositions, including macronutrients (carbohydrates, proteins, fats) and micronutrients (vitamins and minerals). ONSs are classified as “high protein” when they provide ≥20% of energy from protein and “high energy” when they provide ≥ 1.5 kcal/mL according to ESPEN or ≥1.2 kcal/mL according to the Italian legislation [20,34].

Several essential and non-essential compounds with anti-inflammatory properties can be included in the formulation of ONSs. These include omega-3 fatty acids (ω-3), vitamin E, vitamin C, carotenoids, polyphenols, and micronutrients such as selenium and zinc, all of which have been shown to exert beneficial health effects [35]. However, to date, ONSs are currently formulated to treat PEM and do not, instead, consider the different facets of malnutrition and their specific nutritional requirements (e.g., sarcopenia, cachexia, oncological diseases, neurodegenerative diseases, diabetes, etc.). Usually, it is the manufacturer who defines the destination of use and the type of malnutrition of its product. Indeed, there is no comprehensive legislation that clearly and systematically defines the basic nutritional composition of an ONS. In contrast to FSMPs, food supplements in the European Union are regulated as foods [36,37], and do not require medical supervision for their consumption.

## 2. Materials and Methods

Data collection was conducted between June 2023 and July 2023 through face-to-face meetings between consultants from the main companies in the Lombardy region and the clinical team from the Dietetics and Clinical Nutrition Laboratory at the University of Pavia. Additionally, online and paper handbooks were consulted to gather comprehensive information. These handbooks provided details on ingredients, bromatology composition, dosage, description, and indications for use.

### 2.1. Data Construction

The collected data were systematically recorded in an online database (Excel Version 16.83). The database included the following information for each product: energy content (kcal/100 mL, kcal/portion, kcal/mL); protein content (g/100 mL, g/portion), including added amino acids, type of proteins and amino acids composition (Leucine (Leu), Branched-Chain Amino Acids (β-HMB), Branched-Chain Amino Acids (BCAAs ratio), Arginine (Arg), Glutamine (Gln)); lipid content (g/100 mL, g/portion), including types of lipids (Omega-3 fatty acids (ω-3), Eicosapentaenoic Acid (EPA) + Docosahexaenoic Acid (DHA), Medium-Chain Triglycerides (MCT)); carbohydrate content (g/100 mL, g/portion), including types of fiber and glycemic index; micronutrients (calcium, magnesium, iron, zinc, selenium, chromium, vitamin A, D, E, C, B1, B6, B12, and folic acid); texture and flavors available; osmolarity; cost (per 100 mL and per brick); and packaging.

### 2.2. Evaluation Criteria

The adequacy of each product in meeting nutritional indications was evaluated by considering both disease-related malnutrition and inflammation needs [20]. Special attention was given to the most prevalent diseases in the elderly, including cancer, neurodegenerative diseases, diabetes, abdominal surgery, Inflammatory Bowel Disease (IBD), End-Stage Kidney Disease (ESKD), COPD, dysphagia, and age-related anorexia. Moreover, the potential use of FSMPs for treating inflammaging was explored. Diseases characterized by abnormal nutrient absorption through the gastrointestinal tract, which contraindicate oral feeding, were not considered in this analysis.

## 3. Results

Among the 132 products analyzed, 116 were classified as FSMPs, and 16 were classified as supplements, of which 4 were fortified foods. The majority of the products were in liquid form (69 bricks), followed by powder (49 cans/sachets, 19 cans, and 30 sachets) and jars (14, including 10 creams, 3 puddings, and 1 puree). Most of the bricks had a volume of 200 mL (64%) and the majority of the jars (90%) were sold in 125 g portions.

Regarding the taste, the dominant flavors were sweet ones (72%), followed by neutral (26.4%). Most of the products were hyperosmolar (85.7%), followed by hypo-osmolar ones (11.9%). Only two products were iso-osmolar (for further information see Appendix A).

### 3.1. Energy Content

Most of the FSMPs analyzed (among bricks and jars) were hypercaloric (74.7–89.9%), with 15 products (19.0%) providing 400 kcal content per portion. The rest met the ESPEN recommendation to provide at least 400 kcal/day [20] through the intake of 2 portions/day (72.2%) or 3 portions/day (8.8%) (Table 1).

### 3.2. Protein Content

Most FSMPs failed to provide 30 g of protein/portion (98.8%) [20]. Nonetheless, 65.5% of the products achieved this amount with 1 (1.1%), 2 (40.2%), or 3 (24.1%) portions per day, aligning with ESPEN recommendations. The mean grams of protein per portion were 15.8 for bricks and 11.8 for jars. The majority of the products contained only animal proteins (79.5%), with 13.0% providing whey proteins, 10.4% caseins, 39.0% a combination of both, and the remaining products not specifying the composition of the milk proteins (Table 2). Only 1 product contained just plant-based proteins, while 15 products provided both animal and plant-based proteins with the ratio favoring the former. Most products contained soy proteins (738–423.4%), pea proteins (20.0%), or both (6.7%).

### 3.3. Amino Acids Composition

Regarding leucine content, 44 FSMPs reported it with a mean content of 1.5 g/portion (0.47–3.8 g/portion). However, 55.0% of the products did not have an optimal BCAAs ratio [38,39,40,41,42]. Few products contained β-HMB with a mean content of 1.1 g/portion (0.75–1.5 g/portion) (Table 2). Additionally, few FSMPs reported Arg (26.7%) or Gln (21.6%) content.

### 3.4. Fiber and Glycemic Index

Most of the FSMPs (liquid or solid, not powder) did not report Glycemic Index (GI) information, except for those specifically formulated for diabetes (13.9%), which all had a low GI [43]. Examining the fiber content, 40.5% of the products provided both soluble and insoluble fiber, while 43.2% contained just soluble fiber. Fructo-OligoSaccharide (FOS) and/or Galacto-OligoSaccharide (GOS) were the most common (54.0%), followed by inulin (35.1%).

### 3.5. Lipid Content

Of the 75 FSMPs (liquid or solid, not powder), 52 reported ω-3 content (mean 0.86 g/portion, range: 0.07–4.4 g/portion), and 15 indicated EPA plus DHA content (mean = 0.65 g/portion, range 0.1–2 g/portion). Only 21.4% of the 14 products providing MCT proposed them as predominant lipid sources to increase the digestion and absorption of fatty acids [44].

### 3.6. Micronutrients and Cost Analysis

Almost all FSMPs in the form of bricks and jars were complete in micronutrients (vitamins and minerals) in varying amounts. Cost analysis of FSMPs revealed a mean cost per portion of EUR 5.35, with almost half of the products costing less than EUR 5 per portion, and 39.2% costing between EUR 5 and 7 per portion. Specifically, the mean cost per unit of sale for bricks was EUR 6.61 and for jars EUR 6.37, while powders had a lower mean cost of EUR 2.09 per portion. For supplements, the mean cost per portion was EUR 4.17, with most costing less than EUR 5 per portion (for further information see Appendix A).

## 4. Discussion

### 4.1. Descriptive Analysis of the FSMPs Available on the Italian Market

Our analysis showed that most FSMPs on the market meet ESPEN’s recommendations for daily energy and protein daily intake, considering the manufacturer’s indications for use. The majority of products provided adequate leucine content to achieve the desired health benefits. However, there was a notable scarcity of products containing β-HMB, optimal BCAAs ratios, and ω-3. This highlights the need for developing more comprehensive FSMPs that include these components to better support muscle mass and overall health.

Plant vs. Animal Proteins

There was a marked predominance of FSMPs containing animal proteins. While animal proteins have greater anabolic power [45], there is a pressing need to develop products containing a higher proportion of plant proteins to meet sustainability requirements. A combination of both protein sources could still be adequate to meet therapeutic needs [46,47].

Nutrient content and Sustainability

Arg and Gln were present in a few products, and often in amounts that were not sufficient to achieve the desired anti-inflammatory benefits. High doses required for these benefits (around 14–15 g Arg/l; 12 g Gln/l) can typically only be provided by pharmanutrition, far exceeding the possibilities of FSMPs [48].

Although the majority of the FSMPs contained ω-3, the threshold doses for anti-inflammatory effects in older adults in products has not been clearly defined. Soluble fibers were well represented in the analyzed products, with prebiotic functions that positively impact gut microbiota composition [49].

Product Formulation and Consumer Information

There was a lack of salty FSMPs, representing a promising market gap. The intake of savory FSMPs could improve adherence to therapy among elderly patients, as these products would be more similar to those usually consumed, reducing flavor fatigue [50,51,52].

Many products exhibited high osmolarity, which, if consumed improperly (e.g., consumed cold and/or too quickly), could lead to gastrointestinal disturbances. Osmolality plays a crucial role in regulating gastric emptying, intestinal adsorption rates, and the balance of water adsorption/secretion in the small intestine [50]. Specifically, formulations with higher osmolality can cause significant fluid shifts in the stomach and proximal small bowel. This may result in an increased intraluminal volume, accelerating intestinal transit, and placing a greater fluid burden on the large intestine, potentially leading to diarrhea. Such effects are particularly concerning for patients with compromised gastrointestinal function, as they may experience concurrent fluids and electrolytes losses [51]. High osmolality has been identified as a contributing factor to diarrhea frequently observed in patients receiving enteral nutrition [51].

Gradual daily intake of FSMPs (i.e., intake between meals or sips throughout the day) is therefore recommended [52] to prevent premature satiety, which could reduce the intake of regular food [53].

Transparency and Cost

Noteworthy, retrieving comprehensive product information, such as composition, cost, and environmental labels, from official company websites was found to be difficult. It is crucial to provide accurate and detailed information to consumers and clinicians, including nutritional content and packaging sustainability.

The cost analysis highlighted the economic burden of FSMPs, which may be unaffordable for most older adults considering average retirement incomes in Italy [54]. Effective use of FSMPs would likely require local prescription systems for hospital patients, shifting the expense burden to the National Healthcare System [55,56].

### 4.2. Disease-Specific Findings


**Cancer**


Metabolic and nutritional alterations arising during cancer diseases can influence survival, recovery, and the efficacy of treatment in cancer patients, leading to malnutrition, sarcopenia, and cachexia [57]. Malnutrition negatively impacts the quality of life and treatment toxicities, and it is estimated that up to 10–20% of cancer patients die due to the consequences of malnutrition rather than from the tumor itself [57].

As suggested by Muscaritoli et al. 2021 [57] in ESPEN Guidelines for Clinical Nutrition in Cancer, nutritional therapy should preferably be initiated before patients become severely malnourished. In recent years, FSMPs for cancer patients have been introduced to boost immune regulatory functions and delay muscle degradation, especially when an enriched diet is insufficient to meet nutritional goals [57].

FSMPs are often essential for cancer patients who are malnourished or are at nutritional risk, helping to compensate for reduced food intake and prevent nutritional decline during treatment. Specifically, ONSs have been shown to reduce hospital readmissions and length of stay, improve responses to anticancer treatments, and enhance clinical outcomes [58]. These benefits collectively contribute to a better quality of life for patients [59]. Although the total energy expenditure of cancer patients, if not measured individually, is assumed to be similar to healthy subjects (ranging between 25 and 30 kcal/kg/day) [57], nutritional intake could be reduced by symptoms that impact nutrition, thus increasing the risk of malnutrition.

ESPEN guidelines recommend a minimum protein intake of 1.0 g/kg/day and up to 1.5 g/kg/day for cancer patients to help maintain or restore lean body mass [57]. However, many cancer patients do not meet these recommendations due to nutrition-impacting symptoms that affect dietary intake [60].

β-HMB, typically administered at 3 g/day in experimental and clinical research, has been claimed to be an anti-catabolic agent that minimizes protein breakdown [41,42]. A systematic review investigated the effects and safety of β-HMB supplementation in relation to muscle mass, muscle function (i.e., muscle strength and physical performance), and other clinical outcomes (i.e., cancer therapy-related toxicities, hospitalization rate, length of hospital stay, postoperative complications, mortality, and survival) in cancer patients [61]. Six randomized controlled trials (RCTs) were included, involving a total of 416 patients (mean or median age range of 62.0–69.0 years) undergoing treatment for types and stages of cancers. Five RCTs administered β-HMB/Arg/Gln, and β-HMB-enriched FSMPs were provided to patients in one RCT. A beneficial effect of β-HMB supplementation was found on muscle mass and muscle function in the intervention group compared to the control group [61].

Interventions with amino acids have been tested in cancer to optimize nutritional status and counteract muscle mass wasting [62]. These include supplementation with branched-chain amino acids (leucine, isoleucine, and valine), carnitine, and creatine. However, further research is needed to clarify potential benefits.

There is significant interest in ω-3 in cancer due to their potential impact on patients’ outcomes [57]. Two reviews demonstrate that ω-3 improved appetite, body weight, post-surgical morbidity, and quality of life in weight-losing cancer patients [63]. In similar populations undergoing chemo-and/or radiotherapy, ω-3 supplementation showed beneficial effects, most notably in preserving body composition [64]. Despite the known benefits, there is a lack of consensus on supplementation protocols, including dose, duration, formulation (EPA and DHA ratio), and route of administration [65].

In the context of cancer, addressing malnutrition and muscle wasting is critical to improving patient outcomes, as these conditions often arise due to the disease and its treatments. Based on the analyzed products, 66 were indicated for use in the case of malnutrition or increased energy and/or protein needs (61 were FSMPs, 4 supplements, 1 fortified food). Of these, 58 were hypercaloric (≥1.5 kcal/mL or kcal/g), as suggested by ESPEN [20]. Conversely, based on the Italian regulation [34], 62 products were considered hypercaloric (≥1.2 kcal/mL kcal/g). Additionally, 28 products were indicated high in protein content (≥20% energy from protein per portion). Despite this, only 20 products were indicated for use in the case of increased protein needs. β-HMB was present in nine products (seven FSMPs and one supplement) with 3 providing at least 3 g of β-HMB per maximum dosage units/day. Additionally, 55 products were enriched in ω-3, with content ranging from 0.42 g to 9.6 g per maximum dosage units. Among products specifically indicated for malnutrition or increased energy and/or protein needs, 17 contained ω-3.


**Inflammatory bowel disease (IBD)**


IBDs constitute a high-risk condition for malnutrition, especially in the flare-ups of the disease [42]. Protein requirements are increased in the active-stage of IBDs due to inadequate dietary intake, increased rate of protein turnover, and intestinal loss of nutrients or as a result of treatments. Additionally, active inflammation induces a catabolic proteolytic response, warranting intakes of up to 1.2–1.5 g/kg/day in adults. Bowel leakage from diarrhea and inadequate dietary intake from anorexia associated with the disease cause an increased risk of developing micronutrient deficiencies, including iron, calcium, vitamin D, and zinc, which can lead to anemia, growth impairment, and bone fragility [66].

In remission, when energy-protein intakes are insufficient to meet requirements through food consumption, the use of ONSs is an optimal supportive therapy. With the use of ONSs, supplemental intake of up to 600 kcal/day can be achieved without compromising normal food intake in adults [66]. When treatment of the condition requires surgery, the use of oral supplements becomes even more important to optimize preoperative spontaneous oral intake. Indeed, as demonstrated by Kuppinger et al. [67], patients undergoing abdominal surgery with insufficient spontaneous oral intake before hospitalization had a higher rate of postoperative complications. These recommendations apply only when the gastrointestinal tract is still able to absorb nutrients and no intestinal stenosis and/or occlusions are present, which would prevent the use of the gastrointestinal tract itself or give indications of downstream enteral nutrition.

Among the products found in the Italian market, two products were specifically formulated for IBD, three for altered gastrointestinal function, two for increased protein-energy needs but without lipid intake, and three promoted the repair of damaged tissues, the latter being a common condition in chronic inflammatory disease. A total of 21 products could be useful for the active phase of IBD. Of these, two were amino acids supplements. A total of 18 products were hypolipidic or lipid-free, while 2 products contained MCT fats as the main source of fat. Finally, 91% of the products did not contain fiber, and all products were lactose-free.

For the remission phase, 32 products might be indicated (30 FSMPs, 4 supplements, of which 3 were fortified foods). Of these, 21 were lactose-free, including 9 without fiber, and therefore most useful in the early stages of remission, and 12 with fiber, ranging from 0.2 g to 4.95 g per portion, while 4 had lactose and fiber. The majority of the products were hypercaloric (59.3–81.3%) and hyperproteic (68.8%), helping individuals with IBD in remission meet their nutritional needs when they cannot eat enough.


**Chronic obstructive pulmonary disease (COPD)**


COPD is a heterogeneous lung condition characterized by chronic respiratory symptoms due to abnormalities of the airways and/or alveoli that cause persistent, often progressive, airflow obstruction [68]. Elevated blood levels of inflammatory cytokine Tumor Necrosis Factor-α (TNF-α), Nuclear Factor Kappa-β cells (NF-kB), IL-6, and Interferon gamma (INF-γ) contribute to suppressing appetite, increasing energy expenditure, muscle wasting, and promoting cachexia. Malnutrition prevalence varies between 10 and 80% [69,70,71] in the population with this condition. A daily caloric intake equal to 30 kcal/kg/day [20] is recommended, ranging from 25 to 35 kcal/kg/day [72] depending on the Body Mass Index (BMI) of the patients. Regarding amino acids/protein intake, the recommended amount is 0.8–1.5 g/kg for not at risk/no malnourished patients, and up to 1.5 g/kg in patients with sarcopenia [20,72,73]. Several studies have proposed a lower percentage of carbohydrates (about 30%) and a higher percentage of lipids up to 50% (with ω-3 adequate introduction) because it seems more advantageous for ventilatory exchange [74]. Micronutrients commonly deficient in COPD patients include magnesium, copper, selenium, manganese, zinc, folate, and vitamin B12, with vitamin D supplementation highly recommended when deficiency is identified [75,76]. Antioxidant compounds such as ω-3, flavonoids, and some minerals (Zn, Se) from foods seem to exert a beneficial effect [77,78].

In patients with malnutrition, FSMPs should be prescribed [68,72] with a high energy, protein, and micronutrient content in a low volume. National Institute for Health and Care Excellence (NICE) guidelines recommend an average of two ONSs per day in addition to oral intake for at least 12 weeks [72]. Although no specific guidelines exist for patients with COPD, the NOURISH study demonstrated that high protein ONSs can decrease mortality risk and improve handgrip strength, body weight, and nutritional biomarkers within a 90-days post-hospital discharge [73].

Considering all FSMPs and supplements, and considering that COPD leads to an increase of nutrient requirements, only FSMPs and supplements high in energy and protein were considered in the analysis. According to the ESPEN cut-off, 43 products were identified, and 13 (30.2%) of those were balanced in macronutrients (hypercaloric, hyperproteic, lipid range between 20 and 35% of the energy product), and 30 (69.8%) were non-balanced. After removing the balanced products, the percentage of energy per portion from lipids ranged from 37% to 45%, while carbohydrate energy ranged from 35% to 41%. Considering the Italian legislation cut-off, 22 additional FSMPs were identified as hypercaloric, and 4 more products were identified to be suitable for patients with COPD.

Products specifically declared suitable for COPD patients or patients with pneumonia/respiratory diseases were in total seven FSMPs. All (100%) were hypercaloric, and 67% provided a high content of protein. Four products (57%) simultaneously provided a high amount of lipids (>35%) and a low amount of carbohydrates (<45%).

Although further studies are needed to confirm the role of the proportion of carbohydrates and lipids in the diet to minimize unfavorable physiological anomalies in chronic respiratory failure, the composition of these products seems promising for patients with COPD.


**Abdominal surgery**


Malnutrition is frequently observed among patients undergoing abdominal surgery [79] and is associated with poor surgical outcomes, including increased mortality, morbidity, LOS, and readmission rates [80]. Although nutritional therapy for these patients can vary depending on the target organ, the Enhanced Recovery after Surgery (ERAS) protocol was proposed to reduce preoperative stress and improve outcomes.

In the preoperative period, it is mandatory to meet protein requirements, set to 1.2 g/kg body weight/day. For patients at risk of malnutrition before a major surgery, ONSs should be administered for at least 7 days. These oral nutritional supplements should be high-protein (2–3 per day with minimum 18 g protein/dose) and contain arginine, ω-3, and nucleotides, to exert immunonutrition properties [79,81]. This is especially important for those undergoing elective major abdominal surgery, and they should be continued for 7 days post-surgery. Additionally, on the day of surgery, a high protein diet, obtained through ONSs if necessary, should be initiated, except for patients with bowel incontinuity, bowel ischemia, or persistent bowel obstruction. After discharge, high-protein ONSs should be prescribed to meet both energy and protein requirements, especially in previously malnourished, sarcopenic elderly patients.

The ESPEN society strongly endorses the perioperative use of ONSs in surgical patients with malnutrition or at nutritional risk, based on compelling evidence [82,83]. A recent meta-analysis found that preoperative nutritional support with ONSs was associated with a 35% reduction in total complications [21].

Arginine (Arg) has received particular attention for its role in activating T-cell function and serving as a precursor to nitric oxide and proline, both of which are critical mediators of wound healing [79,81]. A systematic review on postoperative outcomes in patients undergoing surgery demonstrated that ONSs enriched with immunonutrients, including arginine, omega 3, and nucleotides, significantly decreased the risk of infectious complications post-surgery and hospital length of stay [84]. Similarly, another systematic review of randomized controlled trials (RCTs) found that perioperative diets supplemented with Arginine, alongside other compounds such as n-3 fatty acids and nucleotides, reduced infectious complications, shortened hospital stays, and reduced total complications from 42% to 27% [85]. It is important to note that these analyses evaluated the effects of multicomponent supplement, as studies on single immunonutrients have not demonstrated comparable effectiveness. This suggests that the combined action of multiple nutrients is essential to fully realize the benefits of immunonutrition [79,86,87]. Additionally, glutamine, produced through the transamination of branched-chain amino acids (BCAAs), plays a key role in supporting immune system activity and protein synthesis, particularly following surgery. However, its clinical benefits in abdominal surgery patients remain inconclusive [79].

Another systematic review and meta-analysis evaluating the impact of post-discharge oral nutritional supplements on outcomes in patients undergoing gastrointestinal surgery also found positive effects, including a reduction in postoperative weight loss and an improvement in biochemical parameters such as serum albumin concentration and hemoglobin in patients receiving ONSs compared to controls [88].

Among the FSMPs found in the Lombardy market, 1 product had specific indication for the Enhanced Recovery After Surgery (ERAS) protocol, 11 were indicated for the postoperative course, 1 for pre-operative preparation, and 2 for perioperative use. Additionally, four products promoted the repair of damaged tissues, including the healing of surgical wounds. Of these 49 products in total, 31 had at least 18 g of protein per dose as indicated by guidelines, and nearly all of the products were hypercaloric (81.6–93.8% depending on whether the criterion was chosen), with an average energy of 333.5 kcal per brick.

Among the products, only 1 was immunomodulatory (containing arginine, ω-3, and RiboNucleic Acid (RNA)), while 7 contained both arginine and ω-3, 1 contained only arginine, and 10 contained only ω-3. For most of the products, an intake of 1 to 3 bricks/day was recommended in combination with normal diet. As a sole source, 4–5 bricks/day were indicated in the ERAS protocol, and it was specified to take 3 bricks for 5 days before major surgery in normonourished patients, and 3 bricks/day for 7 days before surgery in malnourished patients or those at risk of malnutrition.


**Diabetes**


Diabetes is a complex and chronic condition; in its etiopathogenesis, inflammation appears to play a role [89,90]. Glucotoxicity, inflammation, and oxidative stress associated with diabetes lead to decreased skeletal muscle mass, muscle weakness, and impaired physical function. Additionally, diabetes increases the risk of developing comorbidities, which can contribute to malnutrition, sarcopenia, and cachexia, either alone or in combination [91].

Nutrition therapy is pivotal in managing diabetes. Patients should consume large amounts of complex carbohydrates and fiber (at least 35 g per day), include polyunsaturated fats in their diet, and meet protein requirements similar to those of the general population, except in the case of diabetic nephropathy, where a multidisciplinary evaluation is necessary. Regarding micronutrients, the literature is mixed; however, some evidence supports the benefit of chromium, magnesium, and vitamin D. Personalized vitamin B12 supplementation is required when metformin is used [90].

In 2005, the American Diabetes Association stated that short- and long-term use of diabetes-specific formulas (DSFs) as oral supplements is associated with improved glycemic control compared with standard formulas. DSFs are typically higher in fat (40–50% of energy, with a large contribution from monounsaturated fatty acids (MUFAs), e.g., >60% of fat), lower in carbohydrates (∼35–40% of energy), and up to 15% of energy comes from fructose. DSFs are effective in controlling fasting blood glucose and HbA1c, and in increasing High-Density Lipoprotein (HDL) cholesterol [92]. A high proportion of MUFAs also improves metabolic risk factors in patients with diabetes [93].

Among the products on the Italian market, 13 FSMPs and 1 fortified food had a specific indication for diabetes, either in addition to the standard diet (in quantities of 1 to 3 bricks) or on doctor’s suggestion. A total of 17 other FSMPs could also be used as oral supplements for diabetes, as they have adequate nutritional characteristics for this condition. Between 20 and 24 products were hypercaloric, depending on which criterion was used, with an average energy per serving of 272 kcal. Of these, 16 were high-protein and 15 were normoprotein. This wide protein-energy option allows for greater customization, which is very important in diabetes management.

Patients with diabetes may present two opposing phenotypes: (i) patients with type I diabetes are likely to have a weight within the normal range or be underweight, and may benefit from hypercaloric oral nutritional supplementation; (ii) patients with type II diabetes often present with excess weight, corresponding to overweight or obesity, and may require normocaloric but hyperproteic supplementation.

Almost all products contained fiber, with an average of 4.14 g per serving. There was a predominance of soluble fibers, including FOS, acacia gum, inulin, and tapioca dextrins. However, only one product guaranteed the introduction of 30 g of fiber when consumed as recommended [90]. The others provided an average of 11 g of fiber in total, thus not meeting the fiber requirements stated in the guidelines. A total of 14 products reported a glycemic index specification between 13 and 44, with an average value indicating a low glycemic index. A total of 18 products contained ω-3, of which 5 provided 1 g or more per serving, useful for glycemic control and improving the lipid profile of patients with type 2 diabetes [94]. Finally, only one product contained chromium (10 µg/100 mL).


**End-stage kidney disease (ESKD)**


Chronic kidney disease is a state of progressive loss of kidney function, eventually necessitating renal replacement therapy, including dialysis. Up to 50–75% of ESKD patients often face a progressive protein-energy wasting, which adversely impacts morbidity and mortality risk and reduces quality of life [95]. Additionally, inflammation and sarcopenia are often present in these patients [96]. Protein loss due to the passage of blood through the dialysis filter results in increased requirements (1.2 g/kg body weight or higher) in order to maintain a stable nutritional status. Particular attention should be paid to the restriction of certain micronutrients, such as sodium, potassium, phosphorus, and calcium, which can lead to adverse health consequences due to their accumulation in plasma. Finally, of particular importance is limiting fluid intake to avoid excessive interdialytic weight gain [97].

The ESPEN society endorses the provision of ONS to hospitalized patients with acute or chronic kidney disease with or without kidney failure who are malnourished or at risk of malnutrition [98]. The potential clinical benefits indeed include improvements in nutritional status, reduced hospital readmissions, complications, and mortality [98]. Furthermore, ONSs are useful in meeting the nutritional requirements of ESKD patients [98], as they can add up to 10 kcal/kg and 0.3–0.4 g of protein/kg daily over spontaneous intake, favoring the achievement of nutritional goals. Moreover, intradialytic intake of protein-rich foods seems to be effective in mitigating the catabolism associated with the hemodialysis procedure and in increasing total protein intake. Adverse effects related to intradialytic feeding (hypotension, gastrointestinal symptoms, reduced dialysis efficiency, risk of aspiration, and risk of contamination) are not commonly observed and can be avoided with personalized treatment plans [99].

Among the products on the Italian market, 29 FSMPs and 2 fortified foods could be used in ESKD; of those, 3 had specific indications for dialysis. Most of the products were liquid and had a volume between 100 and 220 mL. From the energy-protein point of view, most of the products (71–84%, according to the different classifications) were high in calories, with an average energy of 183 kcal/portion, and high in protein (35%), with an average of 11.2 g/portion. These products had, for 1–3 servings, 168–505 mg of Calcium, 186–557 mg of Potassium, 141–424 mg of Phosphorus, and 96–289 mg of Sodium. These average contents are in line with the need to contain the intake of these minerals in ESKD.


**Neurodegenerative diseases**


Neurodegenerative diseases are characterized by global cognitive impairment, including a decline in memory and at least in one other cognitive domain such as language, visuospatial abilities, or executive function [100]. These conditions have a detrimental impact on nutritional status. Indeed, one-third of older adults with dementia are malnourished, and nearly half are at risk of malnutrition [101]. The mechanisms underlying weight loss in these patients are complex and multifactorial. Dysphagia has been reported in 13–57% patients [102], and can lead to aspiration, a common cause of death in patients with dementia [103].

Several nutrients play important roles in brain integrity and metabolism, including constituents of brain tissue (e.g., fatty acids in neuronal membranes), precursors of neurotransmitters (e.g., specific amino acids), cofactors for metabolic processes (e.g., B vitamins), and antioxidants that protect cells brain from oxidative damage (e.g., vitamin E, selenium, copper). However, ESPEN [100], World Health Organization (WHO) [104], and a recent systematic review by Cochrane [105] do not recommend the systematic use of nutrient supplements to prevent or correct cognitive decline in persons with dementia. However, to ensure an appropriate nutritional status in malnourished people with Alzheimer’s disease and Parkinson’s disease, the ESPEN society recommends the provision of ONSs as part of the medical nutrition therapy to improve energy and protein intake, body weight, well-being, and quality of life [100,106]. Specifically, Tangvik et al. [26] found that FSMPs providing a range from 250 to 850 kcal and 9 to 42 g of protein per day for 7 to 180 days had a beneficial effect on nutritional status in these patients. In cases of specific nutrient deficiencies, the respective nutrients should be supplemented [100]. Therefore, for malnourished patients with dementia, optimal FSMPs should fulfill their higher caloric and protein needs, as well as be suitable for dysphagic patients.

Analyzing the products on the Italian market according to the ESPEN classification [20], 52 FSMPs and 6 supplements were identified simultaneously as hypercaloric (>1.5 kcal/mL) and hyperproteic (≥20% energy from protein per portion). Of those, 33 (63%) had a modified texture (cream, pudding, or thick liquid), suitable for people with dysphagia. Instead, considering Italian legislation [34] (>1.2 kcal/mL and ≥20% energy from protein per portion), 69 FSMPs were identified, of which 21 (30%) had a lipid content between 0 and 1 g/100 mL. Among these, 8 had a creamy texture, 2 had a pudding texture, 22 were powder, and the rest had a liquid texture (46% of the total). Thus, the products ideal for dysphagic patients are well represented. Instead, considering the indication used for neurodegenerative diseases by the producer, nine products were identified. Between those, only four (44.4%) were simultaneously hyperproteic and hypercaloric (>1.5 kcal/mL), and five (55.5%) if considering the Italian legislation (>1.2 kcal/mL). For this reason, a sufficient amount of these products is present to treat malnutrition in this population.

However, new potential bioactive molecules are needed to treat this condition. For example, taurine [107] seems to be a promising compound, but nowadays the evidence is still limited.


**Sarcopenia**


Sarcopenia is a progressive and generalized skeletal muscle disorder associated with an increased likelihood of adverse outcomes including falls, fractures, physical disability, and mortality [108]. Sarcopenia is linked to several risk factors, including malnutrition, immobilization, disease, and inflammation [20,109]. Effective management requires an individualized, multidimensional approach [110] that combines nutritional interventions with physical exercise, which has been shown to be the most effective strategy [108,111]. Protein intake plays a central role in nutritional therapy for sarcopenia [39,108,112], with recommended levels ranging from 1.0 to 2.0 g/kg depending on disease severity and comorbidities [20]. Higher habitual protein intake is associated with lean mass preservation and improved muscle function [113,114], which is particularly relevant for older adults whose muscles are less sensitive to anabolic stimuli compared to younger people [115]. Whey protein, a rich source of highly digestible BCAAs (leucine, isoleucine, and valine) [116,117,118] has demonstrated specific positive effects on intracellular signaling pathways involved in muscle protein synthesis [119]. Among these, leucine has been extensively studied. When combined with proteins and vitamin D, leucine and its metabolite β-HMB (β-hydroxy β-methylbutyrate) appear to offer greater benefits for muscle mass than protein alone [119,120,121,122]. In particular, a multicentre, randomized controlled trial among 380 sarcopenic independent older adults [123] found that supplementation with a product containing whey protein, carbohydrates, fat, vitamin D, trace elements, and 3 g of leucine for 13 weeks significantly improved chair standability [123]. Systematic reviews [124,125] and one umbrella review [39] further suggest that consuming 2–3 g of leucine daily significantly improves muscle mass. Additionally, β-HMB supplementation at an average dose of 3 g/day has been shown to enhance muscle strength [40] and muscle mass [39,40,126] in older adults [39,40,126,127].

Moreover, potential benefits were observed for ω-3 fatty acids. A dose of at least 3000 mg/day DHA combined with more than 800 mg/day of EPA has been associated with positive physical performance in older adults [128,129].

Furthermore, the provision of ONSs for sarcopenic and malnourished older adults is endorsed by the International Clinical Practice Guidelines for Sarcopenia (ICFSR) [130], since it could improve strength outcomes [131].

Analyzing the composition of the FSMPs available in the Italian market according to the dietary sarcopenia needs (protein, leucine, β-HMB, and ω-3), different products were found. First, 68 FSMPs and 6 supplements were hyperproteic (≥20% energy from protein per portion). Of those, 76.5% were hypercaloric (>1.5 kcal/mL) according to ESPEN ranges, while according to Italian legislation, the percentage rises to 92.6% (>1.2 kcal/mL). In contrast, normocaloric products were 23.5% and 7.4%, respectively. Thus, for both legislations, the number of suitable hypercaloric products has a good representation.

Considering all of the products, 32.2% FSMPs and 28.6% supplements reached a leucine content ≥2 g. Specifically, 6 FSMPs and 2 supplements provided this amount in one portion; 31 FSMPs and 1 supplement in two portions; and 1 FSMP and 1 supplement in three portions. A β-HMB content ≥3 g was found only in 6% of all the products: three FSMPs and one supplement. One FSMP and one supplement provided this amount in two portions and two FSMPs in three portions. Therefore, leucine requirements could be reached thanks to several products, while β-HMB was almost absent. EPA and DHA (≥800 mg) were contained in 42 (35.6%) FSMPs and 1 supplement. This amount could be reached mostly with one or two portions; only one FSMP and one supplement provided this amount with three portions. Thus, ω-3 content has an almost sufficient representation.

However, only one FSMP and one supplement contained an adequate amount of protein, leucine, and β-HMB simultaneously, but ω-3 was lacking in these products, indicating the absence of a complete FSMP or supplement for sarcopenia.

According to ESPEN guidelines [75], inadequate or deficit quantities of zinc, selenium, and vitamin D worsen the condition of sarcopenia, and therefore their intake should be monitored. Among products suitable for sarcopenia, 54 (73.0%) contained zinc, 49 (66.2%) contained selenium, while vitamin D was contained in the majority of the products (55, 7%), indicating that micronutrient supplementation is mostly followed.

Of all the products, 16 FSMPs were specifically indicated for sarcopenia by the producer. More than half (56.2%) of these FSMPs were hyperproteic and 81.2% contained an adequate amount of leucine. However, only three FSMPs contained β-HMB, none in an adequate amount. EPA and DHA were adequately present in six FSMPs (37.5%). Regarding micronutrient content, half of the products reported the amounts, with five containing zinc, four selenium, and one vitamin D. None of these products contained all three nutrients simultaneously.


**Dysphagia**


Dysphagia is defined as a condition where the patient has a lower capacity to swallow, experiences difficulty while swallowing food and/or liquids, or is potentially unsafe while swallowing [103]. Malnutrition and dysphagia often occur concurrently, and about 40% of patients with dysphagia were found to be at risk of malnutrition [132]. In these patients, the ESPEN society recommends the prescription of oral nutritional supplements, specifically adapting the texture according to their masticatory and swallowing capacity [22,106]. Specifically, a well-established management strategy for dysphagia is the modification of liquid viscosity by adding a thickening agent in an attempt to reduce the risk of penetration to the airway [133]. Considering texture as the main criterion for identifying suitable products for people with dysphagia, 17 products were identified, and 2 of them were fortified foods. In particular, 10 (58.8%) had a creamy texture, 3 (17.6%) had a pudding texture, 2 (11.8%) were gelling powder, and the remaining products were thicker liquids or purees. Of these, 15 (88.2%) were hypercaloric according to ESPEN and 17 (100%) according to the Italian legislation, of which 11 (64.7%) were high in protein.

Regarding the selected products that reported their indication of the possibility of using the product in case of dysphagia or deglutition difficulties, 13 FSMPs and 3 fortified foods were identified. Most of the products (43.7%) had a cream consistency, three (18.8%) were pudding, two (11.8%) were a gelling powder, and the rest (25%) were pudding, liquid, or puree. In summary, products suitable for dysphagic patients are well represented.

Since the texture of foods is determined by the clinicians according to the characteristics of the patients, it can be useful if the producer indicates different levels of thickness reached by adding certain amounts of the thickening agent.


**Anorexia of aging**


Anorexia of aging is a common geriatric syndrome that includes loss of appetite and/or reduced food intake, and its etiology has been recognized as multifactorial correlating with undernutrition, involuntary weight loss, frailty, and sarcopenia [134].

Since there is a lack of a dedicated guideline on the treatment of anorexia of aging, the ESPEN guidelines on clinical nutrition and hydration in the elderly should be understood as a reference here as well [22]. Nonetheless, a recent systematic review and meta-analysis of randomized controlled trials investigated the effectiveness of ONSs on the main aspects of anorexia of aging. The results of the meta-analysis showed that, generally, ONSs had a positive effect on the overall appetite, energy intake, protein intake, and fat intake. Moreover, the authors found significant differences in the body weight and body mass index between the intervention group and the control group [135].

Among the products on the Italian market, since there are no specific indications for senile anorexia, up to 65 FSMPs and 5 supplements (of which 3 are fortified foods) could be indicated for this condition, including 9 with specific indications from the manufacturer. These products were in the form of liquids, puddings, and creams, to accommodate swallowability. The dominant flavors were sweet ones, in various tastes like vanilla, chocolate, coffee, banana, and berries. In addition, 80–90% of the products were high-energy, according to the different types of classifications, and 66% high-protein, providing valuable nutritional support in just a few servings.

### 4.3. Disease-Related Inflammation

Inflammation is a key component of disease-related malnutrition, significantly contributing to anorexia and impaired nutritional metabolism [3]. It has various effects on metabolism [136], including cytokine-induced anorexia, gastroparesis, and nausea, which are common manifestations of the inflammatory response [136,137].

Additionally, the mobilization of energy stores, caused by the neuroendocrine and inflammatory response, induces the release of fatty acids through lipolysis, the release and breakdown of glucose by glycolysis, glycogenolysis, and gluconeogenesis in the liver, as well as the release of amino acids through muscle proteolysis. These metabolic alterations lead to peripheral insulin, which prevents glucose from entering cells, and so enhances hyperglycemia. The result of all of this is uncontrolled catabolism, which elevates Reactive Oxygen Species (ROS) production and, consequently, inflammation [138].

The release of proinflammatory cytokines, such as IL-6, IL-1β, and TNF-α, is determinant of the pathophysiology of malnutrition. Proinflammatory cytokines impact the brain circuits that regulate food intake, delay stomach emptying, and promote skeletal muscle catabolism [139]. Additionally, there is a connection between gut tissue-released Glucagon-Like Peptide-1 (GLP-1) and proinflammatory cytokines, which leads to decreased food intake and inadvertent weight loss [140].

Research suggests that tailoring nutrition to the specific medical and metabolic condition could further improve the effectiveness of nutrition interventions. Current data indicate that the benefit of nutritional therapy may be predicted by stratifying individuals depending on their inflammatory status (e.g., C-Reactive Protein (CRP) levels) [141]. The EFFORT Study, in fact, revealed that dietary supplementation is less effective in those with severe inflammation (CRP > 100 mg/dl) compared to individuals with CRP concentrations ≤100 mg/L, independently of infection and illness severity [142].

Although the optimal nutritional intervention for addressing varying degrees of inflammation remains unclear, several dietary compounds are known to influence the inflammatory response.

Long-chain fatty acids (LCFAs) serve as substrates for immunomodulatory mediators such as prostaglandins, leukotrienes, and thromboxanes. When derived from the ω-3 family, these mediators exhibit anti-inflammatory properties. ω-3 LCFAs can also competitively reduce the metabolism of ω-6 LCFAs, which produce proinflammatory mediators, as both pathways share the same enzymes. Supplementation with ω-3 LCFAs has been shown to decrease inflammatory markers and improve lean body mass, fat, and skeletal muscle in different types of cancer [143]. However, the evidence is still low to moderate, as reflected in the weak recommendation for ω-3 LCFAs in the ESPEN guidelines on clinical nutrition in cancer [143].

Fiber is another nutrient with anti-inflammatory properties [144]. Indigestible carbohydrates are metabolized by gut microbiota into immune-regulating compounds, such as Short-Chain Fatty Acids (SCFAs), which lower production of cytokines, TNF-α, Monocyte Chemoattractant Protein-1 (MCP-1), or IL-6, thereby attenuating the inflammatory response [145]. Additionally, fiber aids in delivering antioxidants like vitamins or carotenoids to the gastrointestinal tract, supporting a healthy gut microbiota.

Polyphenols also exhibit multiple anti-inflammatory effects, including disruption of immune cell regulation, reduction of proinflammatory cytokine production, and modulation of gene expression [141]. Their antioxidant activity is linked to their ability to scavenge reactive oxygen species (ROS) and bind metal ions. Additionally, approximately 90% of polyphenols reach the large intestine, where they are metabolized by microorganisms into compounds such as SCFAs. These metabolites promote the growth of beneficial bacteria, such as lactobacillus, and positively influence gut microbiota composition [146].

Finally, vitamins and minerals contribute significantly to immune modulation. Long-term supplementation with Vitamins C and E has been shown to enhance immune cell functions in the elderly, including neutrophil phagocytosis and chemotaxis. Similarly, zinc supplementation can increase the proportion of naïve T cells and improve the balance of T helper cells class 1 (Th1) and 2 (Th2) [136]. The term “immunonutrition” refers to the modulation of immune system activity through targeted nutritional interventions [147]. Although there is no standardized approach regarding specific nutrients, timing, or dosage, commonly used immunonutritional formulas include a combination of ω-3 fatty acids, vitamin D, selenium, nucleotides, and sulfur-containing amino acids like glutamine and arginine, provided at supranormal dosages to induce pharmacological effects [147,148]. Immunonutrition or immune-enhanced nutrition has garnered interest, particularly in oncology, and in surgical or critically ill patients [149,150,151,152]. However, ESPEN recommends the use of immunonutrition for cancer patients undergoing surgical treatment of the upper gastrointestinal tract [57], or for malnourished cancer patients undergoing major surgery [153]. Additionally, ESPEN suggests the use of glutamine in critical care, with a potential n-3 FA supplementation in cases of burns and trauma [154].

Therefore, nutrients with anti-inflammatory and/or immunomodulatory properties represent a promising approach to managing disease-associated malnutrition and inflammation. While current evidence remains limited, incorporating such strategies into FSMPs tailored for specific diseases could not only address the underlying condition, but also mitigate associated inflammation.

Among products on the market providing compounds with potential anti-inflammatory effects, the authors identified 55 products containing ω-3, and 87 with Vitamin C, Vitamin E, and zinc. However, only eight products had an indication for the utilization in inflammatory states, such as inflammatory bowel disease, while six had an indication for catabolism/hypercatabolism conditions, and seven for cachexia.

## 5. Conclusions

Our analysis reveals the central role of inflammation in age-related diseases, and underscores that currently available FSMPs do not always effectively address nutritional needs. This review emphasizes the urgent need for products that provide an adequate supply of key nutrients tailored to specific conditions, as illustrated in Figure 1.

To date, no single FSMP fully meets the complex nutritional requirements associated with the polymorbidities often seen in elderly patients (e.g., diabetic nephropathy, sarcopenia with dysphagia). To address this shortcoming, healthcare professionals need to select one or more appropriate FSMP based on the patient’s needs and clinical condition, considering (i) degree of weight loss in malnourished or at-risk patients; (ii) oral nutrient intake; (iii) weight regain goals and timing; (iv) underlying health conditions; (v) bowel function and co-existing diseases; (vi) inflammation levels; (vii) swallowing ability; and (viii) taste preferences. Such a tailored approach is critical to ensuring optimal outcomes for elderly patients with multiple chronic conditions.

Furthermore, as recommended by The Lancet, FSMPs should be prescribed for continuous use for at least one month. After this period, the patient must be re-evaluated to address any issues related to intake (such as taste, mode, and frequency) and to assess the effectiveness of the treatment.

The authors highlight the urgent need to address gaps, such as the identification of specific biomarkers to assess inflammation, and the design of high-quality studies to establish the best nutritional protocols (to be personalized). These protocols should incorporate the targeted use of FSMPs tailored to the patient’s nutritional phenotype, inflammation levels, and comorbidities.

Moreover, the term “FSMPs” was used as an umbrella term in this study, without specifying whether the products were ONSs, as this distinction was often not explicitly declared despite the shared intended use. A clearer definition of ONS could help address this issue in classifying therapeutic dietary products. In light of this, it is recommended to (i) adopt a consistent cut-off such as those used by ESPEN [20], to define hypercaloric and hyperproteic FSMPs, and (ii) to expand the definition of malnutrition beyond the protein-energetic type, to include other forms of malnutrition with clearer indication for nutritional content, based on evidence-supported guidelines (such as at least 3 g/day of leucine for sarcopenic older people). Last but not least, although in some countries, such as Italy, FSMPs can be provided free of charge by the National Health System, the prescription of ONSs remains infrequent and often inappropriate [155]. To effectively combat malnutrition, FSMPs must be made prescribable by a wider range of medical specialists, accessible and affordable for all patients. The economic burden of cost-related malnutrition in the Italian population exceeds EUR 2.5 billion [13]. The use of FSMPs has been shown to significantly decrease hospitalizations, complications, and readmissions, key contributors to healthcare expenditure [21,156]. Therefore, supporting patients through regulations that promote adequate FSMP prescriptions and follow-up monitoring would benefit both individuals and the health care system as a whole [56].

## Figures and Tables

**Figure 1 nutrients-16-04141-f001:**
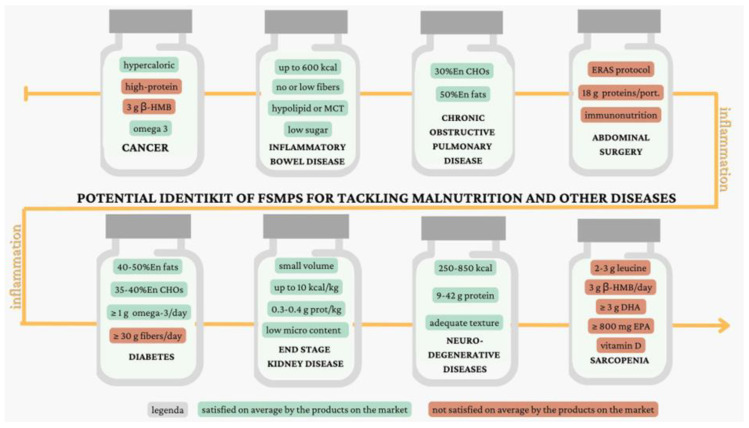
Brief description of the optimal FSMP composition to tackle disease-associated malnutrition. A brief description of the optimal FSMP composition, based on guidelines, systematic reviews, and meta-analysis and/or clinical practice, when evidence was lacking, has been made for malnutrition associated with Cancer [41,42,57,63,64,65]; Inflammatory Bowel Disease [66]; Chronic Obstructive Pulmonary Disease [70,72]; Abdominal Surgery [79,81,87]; Diabetes [90,92,94]; End-Stage Kidney Disease [95,96,97,98,99], Neurodegenerative Disease [100,104,105]; and Sarcopenia [39,40,108,109,112,122,128].

**Table 1 nutrients-16-04141-t001:** Classification of AFMS (*n* = 79, bricks and jars) available on the Italian market based on energy content distinguishing between normocaloric and hypercaloric formulations according to ESPEN guidelines and Italian legislation and their ability to meet the caloric requirements recommended for malnourished individuals. The indications for use were missing for two products, but the authors still assessed whether they reached the threshold value of 400 kcal/day with 1, 2, or 3 servings.

Energy Content	Normocaloric (1–1.19 kcal/mL or kcal/g)	Hypercaloric ESPEN (≥1.2 kcal/mL or kcal/g)	Hypercaloric Italian Legislation (≥1.5 kcal/mL or kcal/g)
Number	7	71	59
%	8.9%	89.9%	74.7%
**Portion needed to provide** **≥400 kcal/day**	**3 portions**	**2 portions**	**1 portion**
Number	7	57	15
%	8.8%	72.2%	19.0%

Note: Abbreviations used: kcal: kilocalories; mL: millilitre; g: gram.

**Table 2 nutrients-16-04141-t002:** Identification of AFMS (*n* and %) available on the Italian market based on protein content, including total protein per portion, the proportion of animal and vegetable protein sources, and the presence of specific amino acids (Leu, valine, and isoleucine). The indications for use were missing for 13 products, but the authors still assessed whether they reached the threshold value of 30 g/day with 2 or 3 servings.

	Type		*n*	%	Total
**Quantity**	<30 g/p		86	98.8%	87
	≥30 g/p [20]	1 p/day	1	1.1%	
		2 p/day	35	40.2%	87
		3 p/day	21	24.1%	
**Animal**	Whey		10	13%	77
	Casein		8	10.4%	
	Whey + casein		30	39%	
	Milk protein		27	35%	
	Milk protein + casein		2	2.6%	
**Plant**	Soy		11	73.4%	15
	Pea		3	20%	
	Both		1	6.7%	
**An vs. Pl**	Just An		62	79.5%	78
	Just Pl		1	1.3%	
	Both		15	19.2%	
**R Leu:Val:Iso**	< 2:1:1		22	55%	40
	≥2:1:1		18	45%	

Note: Abbreviations used: *n*: number; g: gram; p: portion; An: animal; Pl: plant; Leu: leucine; Val: valine; Iso: isoleucine.

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
