# Peer review of "Addressing Inflammaging and Disease-Related Malnutrition: Adequacy of Oral Nutritional Supplements in Clinical Care"

_nutrients, 2024, doi:10.3390/nu16234141_

Round 1
Reviewer 1 Report
Comments and Suggestions for Authors
The article entitled: Addressing Inflammaging and Disease-Related Malnutrition: Adequacy of Oral Nutritional Supplements in Clinical Care has taken into consideration malnutrition in the elder people population. Because their participation as the beneficent of healthcare system service increases not only in the highly developed countries. In the article, authors selected the several health dysfunctions related to age and some diseases which are the hallmarks of older adults. (Cancers, neurodegenerative diseases, inflammatory bowel disease, chronic obstructive pulmonary disease, abdominal surgery, diabetes, end-stage kidney disease, sarcopenia.) Moreover, authors concentrated their attention on supplements and foods for special medical purposes available on the global market. The concentration of proteins, micronutrients, and vitamins in the discussed ONS were considered. Additionally, the attention has been put on the cost of diet supplements. Contrary to other articles malnutrition is the main subject not overweight or obesity. It is an important point for further healthcare studies: life elongation will increase with the number of people in highly developed countries. Therefore, the profiles of diets should be prepared in the same way as pharmacotherapy.
From the editorial point, the article is well-written and readable with well-cited and selected references.
The critical remarks are: introduce to the manuscript the list of used shortcuts with their extensions, extend the captions of the tables by the descriptions of the used parameters. Please make the conclusion part shorter and more synthetic, current it looks like another discussion.
Author Response
The Authors thank and agree with the reviewer for the comments and suggestions.
As suggested to facilitate reading and understanding, a list of abbreviations (and acronyms) has been added following the Abstract (page 2, see “List of abbreviations (and acronyms) and their extensions”) to enhance readability.
As indicated and shared by the Authors, table captions have been expanded to include detailed descriptions in Table 1 (pages 5-6, lines 206-210; 214-215) and in Table 2 (page 6-7, lines 232-237; 241-242).
As request, the conclusion section has been condensed by eliminating digressions and ensuring a more focused summary (pages 19-20, lines 846-916).
Reviewer 2 Report
Comments and Suggestions for Authors
The leucine content or other nutritive compounds are according the information from the label of the product? The content evaluation for this type of products is controlled by a specific laboratory or is controlled just on the producer laboratory?
Author Response
The Authors thank the reviewer for seeking clarification.
The leucine or other nutritional compounds contents reported are based on product labels. These values were not independently verified by the authors and are as stated by the producers.
Reviewer 3 Report
Comments and Suggestions for Authors
Journal Nutrients (ISSN 2072-6643)
Manuscript ID nutrients-3305957
Type Review
Title Addressing Inflammaging and Disease-Related Malnutrition: Adequacy of Oral Nutritional Supplements in Clinical Care
Authors Nagaia Madini * , Alessandra Vincenti , Alice Beretta , Sara Santero , Giulia Viroli , Hellas Cena
The manuscript addresses an important and timely issue in geriatric medicine and public health: disease-related malnutrition (DRM) in older adults, which is often exacerbated by inflammation due to chronic diseases. The paper also explores the role of Foods for Special Medical Purposes (FSMP), including Oral Nutritional Supplements (ONS), in meeting the nutritional needs of this vulnerable population. The focus on FSMPs as a potential solution for DRM is appropriate, given the increasing recognition of nutrition as a critical factor in managing chronic diseases in aging populations. The manuscript is timely and responds to an emerging healthcare priority, namely the malnutrition of older adults.
However, while the manuscript covers an essential topic, the approach and depth of analysis have both strengths and areas that require improvement for a more scientifically rigorous and comprehensive review.
A. Public Health Relevance and Addressing a Gap:
The article highlights an important gap in healthcare for older adults with chronic diseases: the underutilization or inadequacy of FSMPs in meeting the specific nutritional needs of this population. DRM leads to significant negative health outcomes, including increased morbidity, mortality, and healthcare costs. By focusing on FSMPs, the manuscript addresses the role of specialized nutrition in supporting these patients, making the research highly relevant to public health.
B. Diverse Disease Focus:
The manuscript covers a wide range of age-related diseases, such as cancer, neurodegenerative diseases, diabetes, inflammatory bowel disease, end-stage kidney disease, dysphagia, and COPD. This broad scope is beneficial because it encompasses various conditions that frequently lead to malnutrition, offering a comprehensive look at the diverse needs of this population.
C. Adequate Methodology for Data Collection:
The use of face-to-face meetings with company consultants to gather data on 132 FSMPs is a strength, as it provides real-world information on commercially available products. The data collected, including macronutrient and micronutrient content, texture, flavor, osmolarity, cost, and packaging, give a comprehensive picture of the products currently on the market. This approach allows for the identification of trends in FSMP composition and potential shortcomings in addressing patient needs.
D. Identification of Nutrient Gaps:
The manuscript successfully identifies critical nutrient deficiencies in many FSMPs, such as the lack of β-hydroxy-β-methylbutyrate (HMB), optimal ratios of branched-chain amino acids (BCAAs), arginine, glutamine, and omega-3 fatty acids. These nutrients are vital for patients experiencing malnutrition, particularly those undergoing cancer treatment, suffering from sarcopenia, or recovering from surgery. Highlighting these gaps helps to clarify the areas where FSMPs need to be improved.
E. Call for Regulation and Standardization:
The manuscript correctly points out that there is a need for comprehensive regulation and guidelines to standardize the composition of FSMPs. Clear and evidence-based regulations would help ensure that products meet the specific needs of patients with different diseases, improving the consistency and quality of nutritional support.
However, there are some suggestions and comments:
There is an inadequate Discussion of the Role of Inflammation: although the manuscript briefly mentions the emerging role of inflammation in DRM, this topic is not explored in sufficient depth. Inflammation is a key factor that affects nutrient metabolism, alters protein turnover, and influences immune function in patients with chronic diseases. This aspect is critical because FSMPs formulated for inflammatory diseases may need to contain specific nutrients that modulate inflammation, such as omega-3 fatty acids, glutamine, and antioxidants. A more detailed discussion of how inflammation impacts the nutritional needs of patients with diseases like cancer, IBD, and COPD would strengthen the manuscript's relevance to clinical practice. Additionally, a clearer explanation of how FSMPs can address these needs would benefit the reader.
There is a limited Clinical Evidence and Outcome Data: One of the significant shortcomings of the manuscript is the lack of clinical trial data or evidence from observational studies to support the claims made about the efficacy of FSMPs. The manuscript provides data on the nutrient composition of these products but does not present any clinical evidence on how these products impact patient outcomes. For example, data on how FSMPs improve weight gain, muscle mass preservation, immune function, or overall clinical recovery in patients with cancer, COPD, or other diseases would significantly bolster the manuscript's argument. Without such evidence, the claims about the adequacy of FSMPs remain theoretical and may not fully reflect their real-world effectiveness.
The Osmolarity Issue Needs More Depth: The manuscript identifies that a significant proportion (85.7%) of FSMPs have high osmolarity, which can affect patient tolerance. However, the paper does not delve into the potential clinical consequences of high osmolarity, especially for patients with compromised gastrointestinal function, such as those with end-stage kidney disease or dysphagia. A more detailed exploration of how osmolarity impacts product efficacy and patient compliance would be useful, particularly in the context of enteral feeding or in patients who require tube feeding.
There is a lack of Focus on Specific Nutritional Needs for Sarcopenia and Post-Surgical Recovery: Although the manuscript mentions gaps in FSMPs for sarcopenia and abdominal surgery, it does not discuss these areas in sufficient detail. Sarcopenia is a significant concern in older adults, and malnutrition is often a key contributor to muscle wasting and functional decline. The nutritional needs for sarcopenia management (e.g., high-quality protein, HMB, BCAAs) and post-surgical recovery (e.g., arginine, glutamine, vitamin C) are well-established, but the manuscript does not provide a comprehensive analysis of these needs. A more focused discussion on these specific conditions would make the manuscript more clinically relevant.
An issue about cost-Effectiveness and Accessibility:While the article provides the average cost of FSMPs (€5.35 per portion), it does not explore the economic implications of using these products in clinical practice. For example, the cost of FSMPs may be prohibitive for patients in low-income settings, or those without adequate insurance coverage. A more detailed cost-effectiveness analysis, including the potential healthcare savings associated with improved nutrition and reduced hospital readmissions, would provide a more complete picture of the role of FSMPs in disease management.
Regarding the regulatory Recommendations: Although the manuscript calls for comprehensive regulation of FSMPs, it lacks specific recommendations for how such regulation could be implemented. What would the ideal regulatory framework look like? How can current regulatory bodies (such as the FDA, EFSA) incorporate these considerations into existing guidelines for medical foods? Providing concrete suggestions for regulatory reforms would strengthen the manuscript's call to action and offer clearer direction for policy makers and industry stakeholders.
In conclusion: The manuscript provides an important contribution to the field of nutrition in older adults, particularly those with chronic diseases, by addressing the role of FSMPs in managing disease-related malnutrition. It highlights significant nutrient gaps in current FSMP formulations and calls for improved regulation. However, the article would benefit from a more in-depth discussion of the physiological role of inflammation, a deeper exploration of clinical evidence, and a more focused examination of specific disease conditions like sarcopenia and post-surgical recovery. Additionally, the manuscript could be strengthened by discussing the practical challenges related to the cost, accessibility, and regulation of FSMPs.
Overall, while the manuscript offers valuable insights into the adequacy of FSMPs in meeting the nutritional needs of older adults with DRM, it would benefit from greater depth in certain areas, particularly regarding the clinical impact of these products and their cost-effectiveness.
Author Response
There is an inadequate Discussion of the Role of Inflammation: although the manuscript briefly mentions the emerging role of inflammation in DRM, this topic is not explored in sufficient depth. Inflammation is a key factor that affects nutrient metabolism, alters protein turnover, and influences immune function in patients with chronic diseases. This aspect is critical because FSMPs formulated for inflammatory diseases may need to contain specific nutrients that modulate inflammation, such as omega-3 fatty acids, glutamine, and antioxidants. A more detailed discussion of how inflammation impacts the nutritional needs of patients with diseases like cancer, IBD, and COPD would strengthen the manuscript's relevance to clinical practice. Additionally, a clearer explanation of how FSMPs can address these needs would benefit the reader.
The Authors appreciate this important observation. As suggested we have expanded the discussion on the role of inflammation in disease-related malnutrition, highlighting its effects on nutrient metabolism, protein turnover, and immune function (pages 17-18, lines 762-765). The role of specific nutrients with potential anti-inflammatory effects (e.g., omega-3 fatty acids, glutamine, and antioxidants) in prevalent inflammatory diseases has been detailed (pages 17-18, line 792-839).
There is a limited Clinical Evidence and Outcome Data: One of the significant shortcomings of the manuscript is the lack of clinical trial data or evidence from observational studies to support the claims made about the efficacy of FSMPs. The manuscript provides data on the nutrient composition of these products but does not present any clinical evidence on how these products impact patient outcomes. For example, data on how FSMPs improve weight gain, muscle mass preservation, immune function, or overall clinical recovery in patients with cancer, COPD, or other diseases would significantly bolster the manuscript's argument. Without such evidence, the claims about the adequacy of FSMPs remain theoretical and may not fully reflect their real-world effectiveness.
The authors thank the reviewer for highlighting this gap. Thus, the authors have integrated evidence from the ESPEN (European Society for Clinical Nutrition and Metabolism) guidelines, which provide benchmarks for clinical nutrition in Europe. Where disease-specific guidelines are not available, the authors referred to systematic reviews or randomized controlled trials with large sample sizes (page 3, lines 71-80; page 9, lines 329-334; page 11, lines 436-439; page 12, lines 474-477; page 13-14, lines 574-579; page 14, lines 612-614; page 15, lines 666-668; page 16, line 716-719; page 17, lines 740-742 ).
The Osmolarity Issue Needs More Depth: The manuscript identifies that a significant proportion (85.7%) of FSMPs have high osmolarity, which can affect patient tolerance. However, the paper does not delve into the potential clinical consequences of high osmolarity, especially for patients with compromised gastrointestinal function, such as those with end-stage kidney disease or dysphagia. A more detailed exploration of how osmolarity impacts product efficacy and patient compliance would be useful, particularly in the context of enteral feeding or in patients who require tube feeding.
The Authors agree and have added a detailed discussion on the clinical implications of high osmolarity, focusing on patient tolerance and compliance in cases with gastrointestinal sensitivities (page 8, lines 295-304).
There is a lack of Focus on Specific Nutritional Needs for Sarcopenia and Post-Surgical Recovery: Although the manuscript mentions gaps in FSMPs for sarcopenia and abdominal surgery, it does not discuss these areas in sufficient detail. Sarcopenia is a significant concern in older adults, and malnutrition is often a key contributor to muscle wasting and functional decline. The nutritional needs for sarcopenia management (e.g., high-quality protein, HMB, BCAAs) and post-surgical recovery (e.g., arginine, glutamine, vitamin C) are well-established, but the manuscript does not provide a comprehensive analysis of these needs. A more focused discussion on these specific conditions would make the manuscript more clinically relevant.
Thank you for the observation, the authors have enriched the discussion on dietary interventions, emphasizing nutrients for sarcopenia (pages 14-15, lines 637-665) and post-surgical recovery (pages 12, lines 478-494), including high-quality proteins, HMB, arginine, glutamine, and vitamin C.
An issue about cost-Effectiveness and Accessibility: While the article provides the average cost of FSMPs (€5.35 per portion), it does not explore the economic implications of using these products in clinical practice. For example, the cost of FSMPs may be prohibitive for patients in low-income settings, or those without adequate insurance coverage. A more detailed cost-effectiveness analysis, including the potential healthcare savings associated with improved nutrition and reduced hospital readmissions, would provide a more complete picture of the role of FSMPs in disease management.
Thank you for your comment, the authors acknowledge the importance of this issue. While comprehensive cost-effectiveness data in Italy are limited, the authors have included insights from systematic reviews addressing this topic. The Authors also discuss the economic burden of malnutrition related to disease in Italy and the potential savings associated with FSMPs (page 20, line 899-907).
Regarding the regulatory Recommendations: Although the manuscript calls for comprehensive regulation of FSMPs, it lacks specific recommendations for how such regulation could be implemented. What would the ideal regulatory framework look like? How can current regulatory bodies (such as the FDA, EFSA) incorporate these considerations into existing guidelines for medical foods? Providing concrete suggestions for regulatory reforms would strengthen the manuscript's call to action and offer clearer direction for policy makers and industry stakeholders.
Thank you for your useful suggestion. The authors have clarified the need for comprehensive regulation of FSMPs in Italy, emphasizing that current frameworks focus solely on protein-energy malnutrition. The authors propose integrating evidence-based formulations for various malnutrition types into regulatory guidelines (page 20, line 911-916).
Round 2
Reviewer 3 Report
Comments and Suggestions for Authors
Thank you for the review